# Enhanced Storage Performance of PANI and PANI/Graphene Composites Synthesized in Protic Ionic Liquids

**DOI:** 10.3390/ma14154275

**Published:** 2021-07-30

**Authors:** Fatima Al-Zohbi, Fouad Ghamouss, Bruno Schmaltz, Mohamed Abarbri, Mustapha Zaghrioui, François Tran-Van

**Affiliations:** 1Laboratoire de Physico-Chimie des Matériaux et des Electrolytes Pour l’Energie (EA 6299), University of Tours, Parc de Grandmont, 37200 Tours, France; alzohbi-fatima@hotmail.com (F.A.-Z.); bruno.schmaltz@univ-tours.fr (B.S.); mohamed.abarbri@univ-tours.fr (M.A.); 2CNRS, CEA, INSA CVL, GREMAN UMR 7347, University of Tours, IUT de Blois 15 rue de la Chocolaterie, CS 32903, 41029 Blois, France; mustapha.zaghrioui@univ-tours.fr

**Keywords:** polyaniline (PANI), supercapacitors, protic ionic liquids (PIL), graphene oxide (GO), nanocomposites, electrode materials

## Abstract

Polyaniline (PANI) was synthesized using oxidative polymerization in a mixture of water with pyrrolidinium hydrogen sulfate [Pyrr][HSO_4_], which is a protic ionic liquid PIL. The obtained PANI (PANI/PIL) was compared with conventional PANI (PANI/HCl and PANI/HSO_4_) in terms of their morphological, structural, and storage properties. The results demonstrate that the addition of this PIL to a polymerization medium leads to a fiber-like morphology, instead of a spherical-like morphology, of PANI/HSO_4_ or an agglomerated morphology of PANI/HCl. In addition, PAN/PIL exhibits an improvement of the charge transfer kinetic and storage capability in H_2_SO_4_ 1 mol·L^−1^, compared to PANI/HCl. The combination of PANI/PIL and graphene oxide (GO), on the other hand, was investigated by optimizing the PANI/GO weight ratio to achieve the nanocomposite material with the best performance. Our results indicate that the PANI/PIL/GO containing 16 wt% of GO material exhibits a high performance and stability (223 F·g^−1^ at 10 A·g^−1^ in H_2_SO_4_ 1 mol·L^−1^, 4.9 Wh·Kg^−1^, and 3700 W·Kg^−1^ @ 10 A·g^−1^). The obtained results highlight the beneficial role of PIL in building PANI and PANI/GO nanocomposites with excellent performances for supercapacitor applications.

## 1. Introduction

Polyaniline (PANI) has attracted great attention as one of the highest performing conducting polymers for designing novel electrode materials for supercapacitors [1,2,3,4,5,6,7,8,9]. PANI presents a high theoretical specific capacitance (750 F·g^−1^) [10], and more importantly, its popularity is enhanced by its simplicity of synthesis, high thermal stability, and also by the low cost of its monomer [11,12,13]. However, the properties of PANI (morphology, weight mass, and electrical conductivity) and its electrochemical performances considerably depend on the selected synthesis conditions: the pH of the polymerization media [14,15,16,17], concentration of reactants, polymerization temperature and time, stirring conditions [18,19], dopants size and type [2,11,20,21,22,23], oxidants type [24,25,26], etc. Today, there is no debate on whether nanostructured polymers can enhance the electrolyte–electrode contact surface, which can improve the storage capacity and diffusion kinetics in bulk materials. For this reason, scientific research has focused on the synthesis of PANI in the presence of a soft template, such as ionic liquids (ILs) [2,27,28,29,30,31], or a hard template, such as graphene oxide (GO) [32,33,34,35,36] or graphene (Gr) [37,38], in order to grow the PANI nanostructures via non-covalent interactions (hydrogen bonds, π-π stacking interactions, van der Waals force, and electrostatic interaction) as the driving forces [35].

Concerning the “soft template” method, ILs have recently been used as self-assembled molecules in a polymerization medium of aniline [2,28,29]. It should be noted that ILs can be classified into different subclasses, depending on their structure [39]. The two main subclasses are the aprotic ionic liquids (AILs) and the protic ionic liquids (PILs). To date, the chemical polymerization of PANI was solely performed using AIL [2,27,28,29,30,31,40]. Pahovnik et al. have shown that the acidic aqueous solutions of AILs induce the formation of PANI nanostructures during the chemical polymerization of aniline [28,29]. In the same context, Li et al. have shown that the 1-ethyl-3-methylimidalium [emim]^+^-based AIL give spherical PANI particles, instead of a random stacking nano-cudgel morphology, obtained in the absence of AIL [2]. Furthermore, AILs have demonstrated their roles in nanostructuring [2,41] and the enhancement of the capacitance retention of PANI [2]. To the best of our knowledge, no investigation of the synthesis of PANI by oxidative polymerization in a PIL medium has been reported to date, even if PILs are, generally, more easily obtained through a simple proton transfer between an acid and a base of Brönsted than AILs [39].

Regarding the “hard template” method, the combination of a high-surface-area graphene oxide (reduced or not) with PANI can combine the advantages of both materials, therefore providing the possibility to improve the cycle life and energy density of PANI [37,42,43,44,45]. Moreover, it has been reported that the synthesis conditions of PANI/GO nanocomposites, like the mass ratio aniline/GO [37,42,46], the surfactant structure [46,47], etc., greatly influence their electrochemical capacitive performances. The main challenge is to ensure the growth of PANI on the GO surface, which could contribute to the desired synergistic effects [37,42,43,44,45,48]. Li et al. have reported a covalent grafting method for the synthesis of PANI/GO nanocomposites [49]. The functionalization of the surface of the GO with aniline groups via a diazonium reaction provides the covalent grafting of PANI on its surface. The covalently grafted PANI/GO morphology showed a uniform distribution of PANI on the GO surface, compared to the inhomogeneous morphology of non-grafted PANI/GO, leading to PANI/GO sheets and PANI aggregate morphologies. The covalently grafted PANI/GO nanocomposites exhibit a specific capacitance of 442 F·g^−1^ at 0.1 A·g^−1^, which is higher than the values reached using non-grafted PANI/GO (269 F·g^−1^ at 0.1 A·g^−1^) [49]. Recently, ILs have demonstrated their importance in the preparation of nanocomposites by allowing for the dispersion of graphene and GO [50,51] and functionalized GO [52]. However, and to the best of our knowledge, there has been no study on the synthesis of PANI/GO using a PIL medium described in literature to date.

Herein, we investigate the influence of the addition of a PIL into the polymerization medium for the formation of PANI, PANI/Gr, and PANI/GO nanocomposites. Furthermore, PANI and their nanocomposites with GO have been examined in terms of their morphology, chemistry, and electrochemical storage performances for supercapacitor applications.

## 2. Materials and Methods

### 2.1. Materials

Graphite (powder, PROLABO, France), nitric acid (69%, Sigma Aldrich, St. Quentin Fallavier, France), Graphene Gr (Powder, reference: XGnP-H-15, RESCOLL, France), sulfuric acid (95–98%, Alfa Aesar, Paris, France), potassium permanganate (97%, Sigma Aldrich, St. Quentin Fallavier, France), hydrogen peroxide (29–32% ww aqueous Solution, stable Liquid, Alfa Aesar, France), pyrrolidine (≥99%, Merck, France), carbon black (Timcal, super C65, Willebroek, Belgium), activated carbon (Norit, Super, DLC-50, Sigma-Aldrich, St. Quentin Fallavier, France), and poly(tetrafluoroethylene) (PTFE) (60 wt% dispersion in H_2_O, Aldrich, St. Quentin Fallavier, France) were used as received. However, aniline (99.8%, Acros, Geel, Belgium) was distilled before its use.

### 2.2. Preparation of Graphene Oxide

Graphene oxide was prepared from graphite by the Hummer’s method [53], with some modifications, followed by the sonication of the obtained powders. Then, 1 g of graphite was added into a solution containing 6 mL of nitric acid (69%) and 16 mL of sulfuric acid (95–97%). The ingredients were stirred for 15 min, prior to being cooled at 0 °C. Potassium permanganate (4 g) was then added slowly into the suspension (T < 10 °C). The obtained solution was firstly stirred for 2 h at 5 °C and then at room temperature for 30 min. Then, 84 mL of H_2_O was slowly added into the solution (exothermic reaction). Finally, 12 mL of hydrogen peroxide was added to react with the excess MnO_2_. The solution was stirred for 12 h at room temperature. The resulting graphite oxide suspension was filtered and washed with deionized water, until reaching a pH of 7. The final graphene oxide (GO) product was dried under vacuum at 60 °C for 24 h. Attenuated total reflection infrared (ATR-IR) spectroscopy was then used to confirm the GO structure by comparing the ATR-IR spectrum of GO to that of graphite (starting material), as shown in Appendix A of the ESI. The peaks at 3385, 1727, and 1405–1053 cm^−1^ for GO are assigned, according to reference [32], to the O-H stretching, C=O stretching, and C–O in COH/COC (epoxy) functional groups, respectively.

### 2.3. Synthesis of Pyrrolidinium Hydrogen Sulfate [Pyrr][HSO_4_] and Anilinium Hydrogen Sulfate [Ani][HSO_4_]

[Pyrr][HSO_4_] was prepared by an equimolar acid-base reaction between pyrrolidine and sulfuric acid, as reported by our group previously [54,55]. Pyrrolidine was introduced in a three-necked round-bottomed flask immerged in an ice bath and equipped with a reflux condenser, a dropping funnel, and a thermometer. Under vigorous stirring, sulfuric acid was added dropwise to the pyrrolidine, keeping the reaction temperature below 25 °C. The resulting product was dried under primary vacuum (1 Pa) at room temperature for 2 days. The residual water content was then quantified by Karl Fisher coulometer titration (831 KF coulometer) as close to 0.15 wt % as possible. The collected [Pyrr][HSO_4_] obtained with a quantitative yield was then a viscous and pale-yellow liquid at room temperature.

[Ani][HSO_4_] was synthesized by an equimolar reaction between aniline and sulfuric acid. A similar method was employed in preparing [Pyrr][HSO_4_]. A white powder was finally obtained with a quantitative yield.

### 2.4. Synthesis of PANI/GO- and PANI/Gr-Based Materials

PANI/PIL/GO nanocomposites were prepared by the polymerization of [Ani][HSO_4_] in the presence of a suspension of GO in [Pyrr][HSO_4_]. The obtained composites were denoted as PANI/PIL/GO 98/2, PANI/PIL/GO 84/16, and PANI/PIL/GO 68/32, corresponding to a starting material mass ratio of aniline over GO close to 98/2, 84/16, and 68/32, respectively. PANI/PIL/GO 84/16 was prepared as follows: firstly, 0.25 g of graphite oxide was dispersed in [Pyrr[HSO_4_] (63 g, 374 mmol) and treated in an ultrasound bath for 4 h. A stable graphene oxide (GO) dispersion was obtained. Then, anilinium hydrogen sulfate (3 g, 16 mmol) was dissolved in 18 mL of water and added to the suspension of GO. The obtained mixture was cooled at 5 °C. An aqueous solution (9 mL) of ammonium persulfate (APS) (4.5 g, 20 mmol) was slowly mixed into the mixture. The molar ratio of aniline/APS was kept at 1.00/1.25, while the weight ratio of [Pyrr][HSO_4_]/water was maintained at 70/30. The mixture was stirred for 24 h at 5 °C. The resulting product was filtered, washed with deionized water, until reaching a pH of 7, prior to being dried for 12 h at 60 °C under vacuum (1 Pa). For comparison, PANI/PIL/Gr 84/16 (with aniline/Gr 84/16) was also prepared using the same procedure by replacing GO with Gr. PANI/PIL and PANI/HSO_4_ was also prepared using the same procedure in the absence of GO, with and without [Pyrr][HSO_4_] in the polymerization medium, respectively. The conventional PANI (PANI/HCl) was also synthesized by the polymerization of aniline in HCl 1 mol·L^−1^, as described in [54]. The yield of prepared materials was 91–95%. Table 1 summarizes the synthesis conditions of the investigated materials.

### 2.5. Characterizations

Elemental analysis (C, H, N, S and O), performed at Spectropole—Campus Scientifique de Saint Jérôme–Marseille-France, was achieved on a Flash EA 1112 Thermo Finnigan, piloted by Eager 300 software.

Electrical conductivity measurements of the doped PANI samples were conducted at room temperature using a four-probe technique on pellets containing 200 mg of PANI and pressed at 1 ton, with a thickness of 1200–1300 μm, using a keithley 6220 current source and keithley 2182 A nanovoltmetre.

Infra-red characterization was carried out with a Perkin Elmer, Villebon-sur-Yvette, France, spectrum one model (500–2000 cm^−1^) using the attenuated total reflectance (ATR) mode.

Raman measurements on PANI powders were carried out using a micro-Raman spectro-meter (Renishaw^®^, Wottom-Under-Edge, UK), with a laser source of 633 or 514 nm at room temperature.

Scanning electron microscopy (SEM) was performed with a Zeiss ultra plus field emission (Carl Zeiss Microscopy GmbH, Jena, Germany). The powders were deposited on standard SEM aluminum studs, then metallized with platinum (2–4 nm).

### 2.6. Electrochemical Analysis

The electrochemical performances were studied in a symmetric two-electrode cell configuration made from a Teflon Swagelok-type system. The fabrication of electrodes was performed by mixing each obtained composite, carbon black, and PTFE in a mass ratio of 60/32/8 as follows: the composite and carbon black were firstly mixed in a mortar, using ethanol as the solvent. Then, the PTFE was added to obtain a homogeneous paste, which was applied on a glass plate and dried. Finally, for each electrode, ~3 mg of the obtained paste was pressed on stainless steel (0.8 cm of diameter). A porous Whatman membrane (1 cm of diameter) filled with the aqueous sulfuric acid electrolyte (H_2_SO_4_ 1 mol·L^−1^) was used as the separator. Cyclic voltammetry and galvanostatic charge/discharge tests were conducted using a versatile multichannel potentiostat (Biologic S.A), piloted by the EC Lab V10.32 interface, at room temperature. The capacitance values of the electrodes were evaluated from discharge curves using Equation (1) as follows:(1)$$C=\frac{\int Idt}{m∆V}$$
where *C* is the specific capacitance of the active materials of the electrode (F·g^−1^), *I* is the constant discharge current (mA), *d*t** is the discharge time (s), Δ*V* is the voltage difference in the discharge (V), and *m* is the mass of the active material (mg). The energy density and the power density were calculated using the following equation, as described in [48]:
(2)$$E=\frac{1}{2}C\left(∆V^{2}\right)$$
where E, C, ΔV, P, and t are the average energy density (Wh·kg^−1^), specific capacitance (F·g^−1^), potential window of discharge (V), average power density (W·kg^−1^), and discharge time (s), respectively.

## 3. Results and Discussion

In this work, the protic ionic liquid, [Pyrr][HSO_4_], the chemical structure of which is illustrated in Appendix A of ESI, prepared by a simple reaction acid-base, was used as a polymerization medium. [Pyrr][HSO_4_] was mixed with water (weight ratio PIL/water = 70/30) to reduce the viscosity of the neat [Pyrr][HSO_4_].

For clarity, this paper is divided into two parts: the first part describes the impact of adding an PIL to the polymerization medium on the morphology and the properties (such as electrochemical performances) of PANI, while the second part reports on the performances of nanocomposites obtained, owing to the combination of PANI, PIL (as polymerization medium), and GO (as hard template) as new electrodes for supercapacitors.

### 3.1. Polyaniline/Protic Ionic Liquid

In this section, we will investigate how the presence of PIL in the polymerization medium affects PANI properties by comparing the performances of PANI/HCl (conventional PANI), PANI/HSO_4_ (prepared in the absence of [Pyrr][HSO_4_]), and PANI/PIL (prepared in the presence of [Pyrr][HSO_4_]).

To begin with, the morphology of the investigated PANI was studied by scanning electron microscopy (SEM) and is shown in Figure 1. PANI/PIL exhibits a nanofiber morphology (Figure 1A) in comparison with the inhomogeneous and granular morphology of PANI/HSO_4_ (Figure 1B). The granular morphology of PANI/HSO_4_ is consistent with the results described in [56,57,58,59] It should also be noted that PANI/HCl shows an agglomerated morphology (Figure 1C). The difference between the morphology of PANI prepared in water or in an aqueous solution containing [Pyrr][HSO_4_] (PANI/HSO_4_ vs. PANI/PIL) may be related to the kinetic of the nucleation and growth of PANI. There are two modes of PANI nucleation: heterogeneous and homogeneous nucleation [18]. Homogeneous nucleation leads to a nanofiber shape, while heterogeneous nucleation gives rise to granular particles. According to the findings in [2,28,29], the addition of aprotic ionic liquids into a PANI polymerization medium induces homogeneous nucleation and, therefore, a nanostructured PANI morphology by acting as “soft templates”. The nanostructured morphology of PANI/PIL can, therefore, be attributed to the role of [Pyrr][HSO_4_] in controlling the PANI morphology.

The collected CHNS elemental analyses of PANI/HCl, PANI/HSO_4,_ and PANI/PIL are summarized in Table 2. The atomic ratio S/N was calculated to estimate the doping level (degree of protonation) of PANI. The S/N atomic ratio is about 0.15 for PANI/HCl, indicating that PANI/HCl is doped not only with chloride anion, but also with the residual sulfate counter-ions produced by the reduction of the oxidant agent, (NH_4_)_2_S_2_O_8_, during the polymerization. This result is consistent with the results reported in [21,60]. However, in comparison with other PANI materials made during this work, PANI/HCl has the lowest S/N atomic ratio, confirming that it is principally doped with chloride anion. For PANI/HSO_4_ and PANI/PIL, the atomic ratio S/N is 0.2 and 0.3, respectively, indicating an increment in the degree of protonation of PANI, when [Pyrr][HSO_4_] is used as the polymerization medium, as the [HSO_4_]^−^ anion of the PIL can also contribute to the doping of the PANI. For all the investigated samples, the atomic ratio of carbon to nitrogen C/N is 5.8–5.9, indicating no effect of PIL on the PANI backbone.

The electrical conductivity values of investigated materials are also summarized in Table 2. The highest electrically conductivity value, close to 3 S·cm^−1^, was obtained with the PANI/HCl sample, and such a value is comparable with those values already reported in [61,62]. Even if the lowest conductivity values were obtained using polymerization media containing the [HSO_4_]^−^ anion (i.e., PANI/HSO_4_ or PANI/PIL), one can appreciate, based on the data reported in Table 2, that the conductivity of the PANI increases from 0.18 S·cm^−1^ (PANI/HSO_4_) to 1.8 S·cm^−1^ (PANI/PIL), owing to the presence of the PIL. Similarly, the PANI doping level (measured by the S/N atomic ratio) increases from 0.2 to 0.3 by adding the PIL to the medium (PANI/HSO_4_ vs. PANI/PIL). In other words, the doping level of PANI is improved by adding PIL to the polymerization medium, and the electrical conductivity of the resulting PANI is therefore enhanced.

The attenuated total reflection (ATR-FTIR) and Raman spectroscopies were used to investigate the structure of PANI as a function of the polymerization medium. The obtained spectra are shown in Appendix A. The ATR-FTIR spectra of PANI/HCl, PANI/HSO_4_, and PANI/PIL (Appendix A) and the main absorption peaks (Appendix A) indicate the formation of the PANI emeraldine salt form. As can be seen, PANI/HCl and PANI/HSO_4_ exhibit identical ATR-FTIR spectra, which is expected, since the backbone structure of PANI prepared in HCl or in H_2_SO_4_ are identical in each case [63,64,65]. Moreover, the addition of [Pyrr][HSO_4_] to the polymerization medium does not affect the absorption peaks (ATR-FTIR spectrum of PANI/HSO_4_ vs. that of PANI/PIL). Consequently, the presence of PIL does not affect the chemical structure of PANI, which is consistent with the results reported in [28] using Aprotic ionic liquid (AIL), instead of protic ionic liquids (PILs) as the polymerization medium. The Raman spectra of the PANI samples have been recorded to study the possible impact of the addition of PIL to the polymerization medium on the PANI structure. According to the findings in [28], different excitation wavelengths (λ_exc_: 457 (blue), 514 (green), 633 (red), and 1064 nm (near infra-red)) can be used to examine the molecular structure of PANI. It is important to note that stretching the vibration of benzenoïd and quinoïd rings depends on excitation lines [66]. The Raman spectra of the investigated materials obtained using 633 nm or 514 nm radiation excitations are shown in Appendix A. The PANI main bands were assessed, according to the procedure described in [28,67], as follows: 1591 cm^−1^ C-C stretching of the benzenoid units, 1500 cm^−1^ C=C stretching of the quinoid ring, 1378 and 1338 cm^−1^ C-N^+●^ stretching in polarons, 1256 cm^−1^ C=C stretching of benzene diamine units, 1168 cm^−1^ C-H in plane vibration of C-H, 808 cm^−1^ in plane vibration of the benzenoid ring and out of plane vibration of C-H, and 514 cm^−1^ torsion of C-N-C and out of plane C-N vibration. As shown in Appendix A, the spectrum obtained, with a radiation of 633 nm of PANI/PIL, is almost identical to that of PANI/HSO_4_. The Raman spectra are in agreement with the ATR-FTR results, confirming that PANI is obtained in its conducting form. Therefore, the presence of PIL in the polymerization medium does not induce the formation of an undesirable structure of PANI, such as phenazine, as already observed for other specific aqueous media [11]. Moreover, the Raman spectra obtained with a 514 nm excitation wavelength (Appendix A) show a red shift to 545, 836, and 1645 cm^−1^ from 514, 812, and 1618 cm^−1^, respectively. Some studies have showed that the doping level [68,69], dopant structure [68,69], and polymerization medium of PANI [70,71] could influence the Raman peak position. Based on the elemental analysis, showing the increment of the PANI doping level when [Pyrr][HSO_4_] is used in the polymerization medium, one can assume that this Raman peak shift could be due to this doping level. A red shift of the polaronic band from 1338 to 1352 cm^−1^ can also be noticed, which could be due to a specific interaction between the PIL and the cation radical of the doped PANI backbone. In the following, the electrochemical performances of PANI/PIL will be compared with those of PANI/HCl, which exhibit the highest electrical conductivity among all prepared samples (Table 2).

The electrochemical performances of PANI/PIL and PANI/HCl were studied in H_2_SO_4_ 1 mol·L^−1^ in a symmetric two-electrode configuration.

Figure 2 presents the CV curves recorded at scan rates of 15 mV·s^−1^ in a potential window of 0.8 V. Both CV curves show a typical quasi-rectangular shape with a redox peak, highlighting the pseudocapacitive behavior of the Pani electrodes [28,29,32]. The CV curves exhibit a rectangular shape with a more pronounced deviation for Pani/HCl at the beginning of the forward and back scans, which is related to electrode polarization. PANI/PIL exhibits a more rectangular shape of the CV curve than PANI/HCl, indicating a lower polarization probably due to a lower charge transfer resistance. As is commonly reported, the electrochemical performances of PANI are related to its morphology [58,72,73], as well as to the redox reactions involving counter-ions doping/de-doping from PANI. The quasi-rectangular shape of the PANI/PIL CV curve can thus be attributed to its nanostructure and fibrillary morphology.

### 3.2. PANI/PIL/GO and PANI/PIL/Gr Nanocomposites

As we reported in the previous section, [Pyrr][HSO_4_], could act as a “soft template” in the polymerization medium, thus generating a fiber-like morphology of PANI. The resulting material showed a lower polarization resistance and high electrical conductivity. In this section, we will present and discuss the impact of the combination of [Pyrr][HSO_4_] (soft template) and Graphene oxide (GO) on the electrochemical performances of PANI. The synthesis scheme is illustrated in Figure 3. The PANI/PIL/GO nanocomposite was synthesized in the presence of GO and by following the same synthesis procedure of PANI/PIL. Three different aniline/GO mass ratios (98/2, 84/16, and 68/32) were investigated. The obtained nanocomposites are denoted as PANI/PIL/GO 98/2, PANI/PIL/GO 84/16, and PANI/PIL/GO 68/32. In addition to the aniline/GO mass ratio, the effect of surface function groups was studied by replacing GO by reduced graphene oxide (Gr) during the synthesis. Therefore, an aniline/Gr nanocomposite with a mass ratio of 84/16 was prepared (PANI/PIL/GO 84/16 vs. PANI/PIL/Gr 84/16). Indeed, GO can be considered as a functionalized Graphene with oxygen functional groups (epoxide, hydroxyl, carboxyl, etc.) [74], as illustrated in Appendix A. The successful polymerization has been validated by characterizing PANI/PIL and all the investigated nanocomposites by ATR-IR spectroscopy. All the obtained spectra are shown in the Appendix A.

Figure 4 presents SEM images of the PANI/PIL/GO with different aniline/GO mass ratios and the PANI/PIL/Gr 84/16 nanocomposites. It is clear that the Graphene particles have been uniformly coated by PANI. According to the literature, both Gr and GO can interact with Anilinium (phenyl group) by π-π stacking interactions, promoting the polymerization of aniline on the surface of Gr or GO [49,75]. However, only GO, owing to its oxygen functional groups, can also interact through hydrogen bond interactions with PANI [48]. PANI homogenously covers the whole surface of the GO or Gr, except for PANI/PIL/GO 68/32, as uncovered GO sheets are observed (C in insert), which indicates that a higher level of PANI is required for a full recovery of the GO layers.

The electrical conductivity (σ) of the investigated nanocomposites are as follows: 1.18 S·cm^−1^ for PANI/PIL, 1.4 S·cm^−1^ for PANI/PIL/GO 84/16, 0.4 S·cm^−1^ for PANI/PIL/GO 69/32, and 13 S·cm^−1^ for PANI/PIL/Gr 84/16. It appears that although the limited electrical conductivity of GO (~10^−3^ S·cm^−1^) was caused by its surface functions [35], a low rate of GO does not significantly impact the conductivity of nanocomposites, since the PANI/PIL/GO 84/16 electrical conductivity (1.4 S·cm^−1^) is very close to that of PANI/PIL (1.18 S·cm^−1^). This result indicates that the collection of charge through PANI would still be effective in PANI/PIL/GO 84/16. However, an increase of the GO composition induces a clear decrease of the nanocomposite conductivity. For example, the conductivity value of PANI/PIL/GO 68/32 (0.4 S·cm^−1^) is three times lower than that of PANI/PIL. Furthermore, a clear enhancement of the conductivity is observed by substituting the GO by Gr. For example, the electrical conductivity of PANI/PIL/Gr is close to 13 S·cm^−1^, which is one order of magnitude higher than that of PANI/PIL/GO 84/16. This difference in electrical conductivity between these two materials is attributed to the higher electrical conductivity of reduced graphene Gr, compared to GO [76].

The electrochemical storage performance of all the investigated materials was studied in H_2_SO_4_ 1 mol·L^−1^ using a symmetrical two-electrode configuration. The symmetrical devices have been cycled in a voltage range of 0.8 V. Figure 5a,b present the recorded cyclic voltamograms (CV) at 15 mV·s^−1^ using the investigated nanocomposites. All CV curves exhibit a clear redox peak assigned to the doping-de-doping reaction of PANI. After comparing all the investigated nanocomposites, it appears that the CV curve of the PANI/PIL/GO 84/16 is more capacitive and the redox system is more pronounced than those observed for the other investigated electrodes. In other words, the electrochemical performances of the PANI/GO nanocomposites are related to the aniline/GO mass ratio, as already reported in [37,42,46]. A similar conclusion was reported by Li et al., who reported that the best electrochemical behaviors are obtained for PANI/GO prepared with an aniline/GO ratio close to 84/16 [49].

Figure 5 shows the specific capacitances of the investigated materials measured during the discharge of the galvanostatic charge/discharge (GCD) curves. We can notice that, for all the studied current densities, the specific capacitance of PANI/PIL/GO 84/16 is systematically higher, compared to the other investigated materials. More interestingly, increasing the current density has a limited impact on this composite, while a drastic decrease of the specific capacitance is observed in Pani/HCl. Figure 4D shows an example of GCD curves obtained with the investigated materials. First of all, the Pani materials prepared in the protic ionic liquid exhibit higher charging and discharging times, as well as a smaller ohmic drop. Adding GO leads to a greater enhancement.

The energy and power densities at different current densities of PANI/PIL, PANI/PIL/GO 84/16, and PANI/HCl are shown in the Ragone plot in Figure 6 (blue symbols) and Appendix A. At 1 A·g^−1^, the specific energy density is close to 5.6, 6.4, and 6.7 Wh·kg^−1^ for PANI/HCl, PANI/PIL, and PANI/PIL/GO 84/16, respectively. The specific energy densities decrease to 1.5, 3.8, and 5 Wh·kg^−1^, respectively, by increasing the current density from 1 to 10 A·g^−1^. According to Figure 6, one can see that the specific energy densities of the PANI/PIL/GO 84/16 nanocomposite is higher than that of PANI/PIL, demonstrating the enhancement of the specific energy of device based on PANI/PIL through the addition of GO. Under the studied conditions, PANI/PIL/GO has a maximal specific energy of 6.7 Wh·kg^−1^ and a maximal specific power of 3700 W·kg^−1^ at 0.7 and 10 A·g^−1^.

To show the importance of the addition of the PIL to the polymerization medium, the performance of our materials were compared with those of covalently grafted PANI/GO nanocomposites and is already reported in [49]. The energy and power densities at 10 A·g^−1^ of the covalently grafted PANI/GO and the nongrafted PANI/GO taken from the findings in [49] are presented in the Ragone plot in Figure 6 (red symbols). Prior to directly comparing their performances with those obtained during this work, we report the synthesis and measurement conditions reported in the literature for each material below [52]. The covalently grafted PANI/GO was synthesized by a three-step synthesis from graphite flakes and an aniline monomer. Firstly, GO was prepared by a modified Hummer’s method. Then, aniline groups were grafted on the surface of the obtained GO via a diazonium reaction. Finally, a covalently grafted PANI/GO nanocomposite (aniline/GO: 86/14 mass ratio) was prepared by an in situ polymerization, using HCl as the polymerization medium and ammonium persulfate as the oxidant agent [49]. For comparison, the nongrafted PANI/GO nanocomposite was synthesized in the presence of pristine GO [49]. The electrochemical performances of these materials were performed in H_2_SO_4_ 1 mol·L^−1^ between 0 and 0.8 V using a two-electrode configuration [49]. The typical mass loading of the nanocomposite/super P/Polyvinylidene fluoride 80/10/10 on each electrode was about 1 mg·cm^−2^. From Figure 6 and according to the findings in [49], one of the first things to note is that covalently grafted PANI on GO enhances the power density of the nanocomposite PANI/GO: it increases from 3160 Wh·Kg^−1^ for non-grafted PANI/GO to 3500 Wh·kg^−1^ for covalently grafted PANI/GO at 10 A·g^−1^. Moving to the other main point in the Ragone plot, it is noticeable that the energy and power densities of the covalently grafted PANI/GO taken from the findings in [49] (4.8 Wh·kg^−1^ and 3500 W·kg^−1^ @ 10 A·g^−1^) are very close to those obtained here using the PANI/PIL/GO 84/16 (4.9 Wh·kg^−1^ and 3700 W·kg^−1^ @ 10 A·g^−1^). In other words, during this work, we were able to obtain nanocomposites with good performances (specific energy and specific power), without resorting to the intermediate step of the aniline functionalization of GO, as reported in [52] and, more importantly, with an electrode loading (3 mg·cm^−2^) three times higher than that used in [52].

It should be noted that, according to the findings in [37,42,43,44,45,48] and as shown in Appendix A, both GO and Gr exhibit a low specific capacitance due to their tendency to be aggregated through π-π interactions. However, the combination of Gr or GO with PANI enhances the specific capacitance of the resulting material, owing to the synergistic effect between Gr or GO and PANI [37,48]. The PANI growth on the Gr surface inhibits the stacking/agglomerating of GNS (enhanced electrode/electrolyte interface areas) and Gr in the composites provides a highly conductive path for electron transport during the charge/discharge processes. It should also be noted that the presence of [Pyrr][HSO_4_] in the polymerization medium has positive effects on the specific capacitance of PANI (173 F·g^−1^ for PANI/PIL and 66 F·g^−1^ for PANI/HCl at 10 A·g^−1^). [Pyrr][HSO_4_] can interact with anilinium salt and with GO through electrostatic interactions and control the growth of PANI on the GO surface (PIL controls the PANI morphology, as illustrated in Figure 1). Therefore, the excellent electrochemical behavior of PANI/PIL/GO 84/16 can be explained by the synergistic effect of the three components (GO, PANI, and [Pyrr][HSO_4_]).

The cycling stability of PANI/PIL/GO and PANI/PIL/Gr was studied by consecutive galvanostatic charge–discharge measurements at 2 A·g^−1^. Figure 7 shows the capacitance retention for 1000 charge/discharge cycles. As we can see, all the materials exhibit a capacity retention exceeding 60% after 1000 cycles. However, the capacitance retention of the PANI/PIL/GO is related to the aniline/GO mass ratio. The capacitance retention of PANI/PIL/GO 98/2 and PANI/PIL/GO 84/16 are close to 63% and 79%, respectively, indicating that a relatively high GO composition can improve the cycling stability of PANI/PIL/GO. Moreover, the capacitance retention of PANI/PIL/GO 84/16 (79%) is higher than that of PANI/PIL/Gr (67%), showing the beneficial impact of GO vs. Gr. In other words, the hard template type can play a key role in the enhancement of PANI electrochemical performances.

## 4. Conclusions

In summary, we have reported the effect of the combination of protic ionic liquid [Pyrr][HSO_4_] (soft template), graphene or graphene oxide (hard template), and PANI on the electrochemical performances of the resulting materials (denoted PANI/PIL/Gr or PANI/PIL/GO).

On the one hand, we have shown that [Pyrr][HSO_4_] can control the PANI morphology and generate fibrillar PANI particles in comparison with the agglomerated morphology observed in the case of the conventional PANI (PANI/HCl). In addition, PANI prepared in the presence of [Pyrr][HSO_4_] is less resistive and more capacitive than PANI/HCl-based electrode materials for supercapacitors. On the other hand, the electrochemical performances of the nanocomposites, PANI/PIL/GO, depend on the aniline/GO mass ratio, as well as the hard template type (GO or Gr). The best performances are obtained with Pan/PIL/GO 84/16, which is also more effective than the nanocomposite based on graphene (PANI/PIL/Gr 84/16). PANI/PIL/GO 84/16, prepared with the aniline/GO mass ratio of 84/16, exhibits a high specific capacitance (223 F·g^−1^ @ 10 A·g^−1^) and large energy and power densities at a high current intensity (4.9 Wh·kg^−1^ and 3700 W·kg^−1^ @ 10 A·g^−1^), thanks to the synergistic effect of GO as a hard template and PANI and [Pyrr][HSO_4_] acting as a soft template.

To conclude, the presence of PIL in a polymerization medium induces the formation of efficient PANI/GO nanocomposite-based electrode materials for supercapacitors.

## Figures and Tables

**Figure 1 materials-14-04275-f001:**
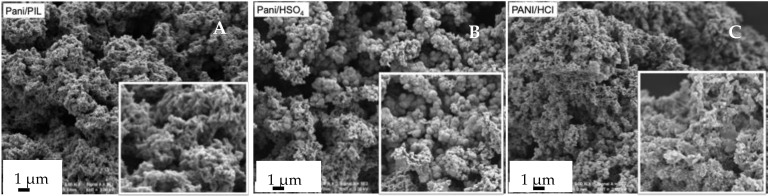
SEM images of PANI/PIL (**A**), PANI/HSO_4_ (**B**) and PANI/HCl (**C**).

**Figure 2 materials-14-04275-f002:**
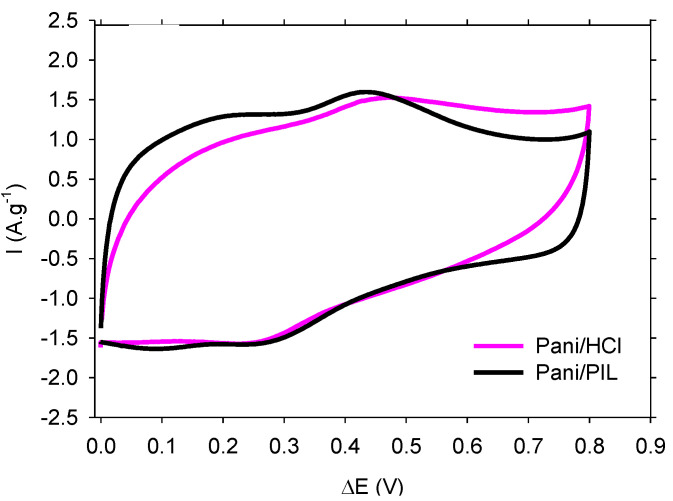
CV curves at a scan rate of 15 mV·s^−1^ of PANI/HCl and PANI/PIL electrodes in H_2_SO_4_ 1 mol·L^−1^.

**Figure 3 materials-14-04275-f003:**
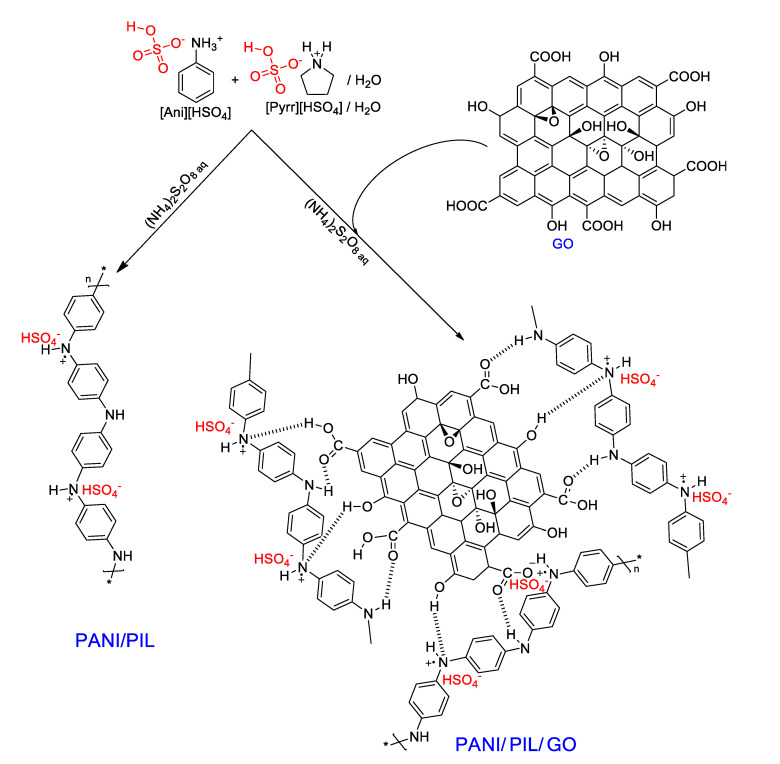
Schematic illustration of the synthesis of PANI/PIL and PANI/PIL/GO composites.

**Figure 4 materials-14-04275-f004:**
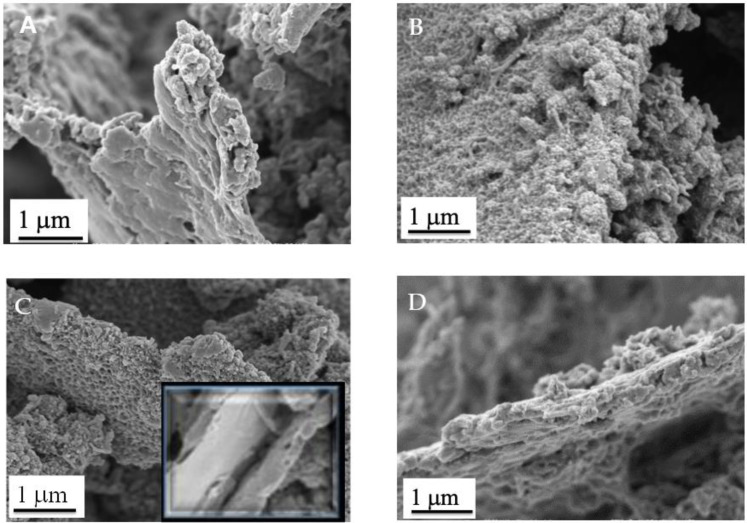
SEM images of PANI/PIL/GO 98/2 (**A**), PANI/PIL/GO 84/16 (**B**), PANI/PIL/GO 68/32 (**C**), and PANI/PIL/Gr 84/16 (**D**).

**Figure 5 materials-14-04275-f005:**
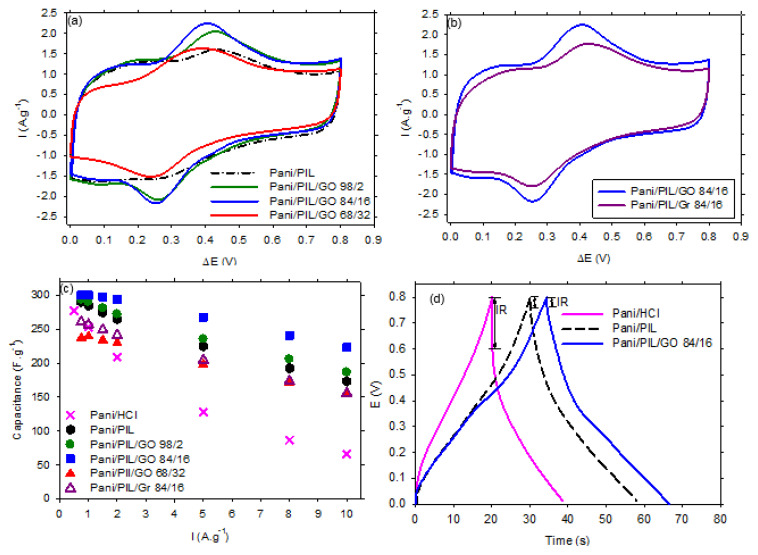
Cyclic voltamograms at 15 mV·s^−1^ of (**a**) PANI/PIL and PANI/PIL/GO (98/2, 84/16 and 68/32) and (**b**) PANI/PIL/GO 84/16 and PANI/PIL/Gr 84/16 nanocomposites in H_2_SO_4_ 1 mol·L^−^^1^, (**c**) the specific capacitance at different current densities of all the investigated electrodes, and (**d**) the charge discharge curves at 2 A·g^−1^ of PANI/HCl, PANI/PIL, and PANI/PIL/GO 84/16.

**Figure 6 materials-14-04275-f006:**
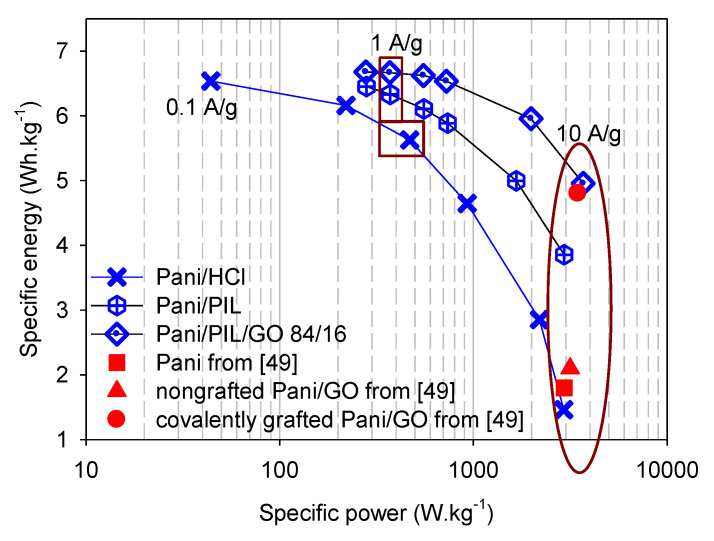
Ragone plot of the investigated materials vs. the materials studied in [49] in H_2_SO_4_ 1 mol·L^−1^.

**Figure 7 materials-14-04275-f007:**
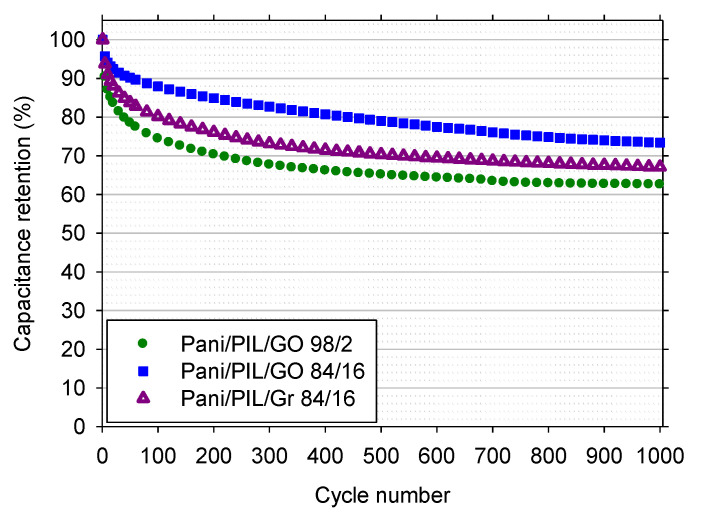
Cycling stability at 2 A·g^−1^ of PANI/PIL/GO 98/2 and PANI/PIL/Gr 84/16 nanocomposites in H_2_SO_4_ 1 mol·L^−1^.

**Table 1 materials-14-04275-t001:** Reactant, polymerization medium, and aniline/GO mass ratio of different investigated PANI.

Product	Reactant	Polymerization Medium	Gr or GO	Aniline/GO or Aniline/Gr
PANI/HCl	aniline	HCl 1 mol·L^−1^	–	100/0
PANI/HSO_4_	[Ani][HSO_4_]	water	–	100/0
PANI/PIL	[Ani][HSO_4_]	[Pyrr][HSO_4_]/water (70/30)	–	100/0
PANI/PIL/GO 98/2	[Ani][HSO_4_]	[Pyrr][HSO_4_]/water (70/30)	GO	98/2
PANI/PIL/GO 84/16	[Ani][HSO_4_]	[Pyrr][HSO_4_]/water (70/30)	GO	84/16
PANI/PIL/GO 68/32	[Ani][HSO_4_]	[Pyrr][HSO_4_]/water (70/30)	GO	68/32
PANI/PIL/Gr 84/16	[Ani][HSO_4_]	[Pyrr][HSO_4_]/water (70/30)	Gr	84/16

**Table 2 materials-14-04275-t002:** Elemental composition and electrical conductivity (σ) of the investigated PANI.

*Sample*	Percentage (%)				Atomic Ratio		σ
	C	H	N	S	C/N	S/N	S·cm^−1^
PANI/HCl	55.6	4.7	11.1	3.7	5.8	0.15	3
PANI/HSO_4_	52.3	4.6	10.3	5.4	5.9	0.2	0.18
PANI/PIL	51.1	4.6	10.2	7.1	5.8	0.3	1.18

## Data Availability

The data presented in this study are available on request from the corresponding author.

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
