# Peer review of "Enhanced Storage Performance of PANI and PANI/Graphene Composites Synthesized in Protic Ionic Liquids"

_materials, 2021, doi:10.3390/ma14154275_

Round 1
Reviewer 1 Report
The author investigated a protic ionic liquid, as s soft template, to synthesize PANI for the application of supercapacitor. The result showed that the ionic liquid impact the morphology of the obtained materials. PANI electrodes, prepared by the ionic liquid, exhibited higher capacitance, and improved energy and power densities. The results are interesting. Some concerns need to be addressed before further consideration.
- Figure 1 was not mentioned in the manuscript. It is not clear how the role of ionic liquid over the synthesis of composite materials. The mechanism of PyrrHSO4 needs to be discussed.
- The electronic conductivity of prepared PANI needs to be measured using either the four-probe or EIS method.
- It is strongly suggested to prepare a table to benchmark the performance of obtained material, regarding specific capacitance and energy/power performance.
Author Response
On behalf of all the authors, I hereby submit our revised manuscript entitled “Enhanced storage performance of PANI and PANI/graphene composites synthesized in protic ionic liquids” . Below our response to reviewer comments (reviewer 1, 2 and 3). Changes made to the paper, following to the reviewer’s recommendations, are highlighted with yellow color. The response are in the attached word file

Reviewer 2 Report
I write you in regards to the manuscript entitled “Enhanced storage performance of PANI and PANI/graphene composites synthesized in protic ionic liquids”, submitted to the Materials. After going through the manuscript, I have got the belief that it is written in a suitable manner and the content could be interesting for the researchers in this field in term of fabricating PANI and PANI/graphene composites synthesized in protic ionic liquids and employed as electrodes for the supercapacitor. The manuscript was well-prepared with scientific sound and logic. However, there are some aspects where more work is requested. For the guidance of the authors, these are included below. Please consider them and deliver a revised version for reconsideration for publication.
- The scale bar in the SEM image (figure 1) should be enlarged for the better vision.
- Labels of Figure 4 is missing. Please check and correct.
- There are many graphene/PANI nanocomposite for supercapacitor reported in the literature. It is necessary to compare the performance of the PANI/graphene as electrode for the supercapacitor in this work to others in the literature.
- There are several English errors throughout the manuscript such as line 123: “synthesis” should be “synthesized”.
Author Response
On behalf of all the authors, I hereby submit our revised manuscript entitled “Enhanced storage performance of PANI and PANI/graphene composites synthesized in protic ionic liquids” . Below our response to reviewer comments (reviewer 1, 2 and 3). Changes made to the paper, following to the reviewer’s recommendations, are highlighted with yellow color. Repsonse are in the attached word file.

Reviewer 3 Report
In their manuscript, the authors report a novel synthesis route for polyaniline (PANI) relying on protic ionic liquids (PILs). PANI is a conducting polymer which can be oxidized with a theoretically high capacity, making it a promising material for energy storage devices such as supercapacitors. However, the actual capacity of PANI strongly depends on its nanostructure, which is essentially governed by the synthesis conditions.
The key finding of the authors is that by using a PIL, the nanostructure of PANI can be tuned to improve its electrochemical performance. In particular, when polymerized in the presence of a PIL (soft template), PANI exhibits a fibrous structure as well as an improved conductivity and capacity as compared to conventional synthesis routes.
Additionally, the authors use graphite or graphene oxide particles (so-called hard templates) in the PIL-based synthesis of PANI to direct the polymerization via surface interactions with the template, yielding nanocomposite materials which show improved electrochemical properties (conductivity and capacity) as in the soft template case.
As far as I am aware, only aprotic ILs have been used for the synthesis of PANI so far, that is, the approach presented by the authors constitutes a novel route to synthesize promising energy storage materials. Furthermore, the manuscript is clearly written and the findings reported therein are supported by the data presented. Therefore, I believe that this manuscript will certainly be interesting for many readers of Materials.
The only minor flaws I found are as follows:
There seems to be an issue with the figure labeling and referencing in the text, especially with regard to the ESI. Moreover, in the methods part the authors mention DFT calculations, however, these results are not presented or discussed.
For these reasons, I recommend publication of this manuscript after addressing these issues.

Author Response

(The authors gave the same response as above.)

Round 2
Reviewer 1 Report
All my concerns have been addressed. I recommend it for publication.